# Analysis of Nitrogen Uptake in Winter Wheat Using Sensor and Satellite Data for Site-Specific Fertilization

**Matthias Stettmer** [1,*] , **Franz-Xaver Maidl** [2], **Jürgen Schwarzensteiner** [3], **Kurt-Jürgen Hülsbergen** [2] **and Heinz Bernhardt** [1]

1   Agricultural Systems Engineering, TUM School of Life Sciences, Technical University of Munich, Dürnast 10, 85354 Freising, Germany; heinz.bernhardt@tum.de
2   Organic Agriculture and Agronomy, TUM School of Life Sciences, Technical University of Munich, Liesel-Beckmann-Straße 2, 85354 Freising, Germany; maidl@tum.de (F.-X.M.); kurt.juergen.huelsbergen@tum.de (K.-J.H.)
3   Farmtastic Consulting GmbH, Graf-von-Bray-Straße 14, 94342 Irlbach, Germany; js@farmtastic.consulting
*   Correspondence: matthias.stettmer@tum.de

**Abstract:** Sensor- and satellite-based determination of nitrogen uptake provides critical data in site-specific fertilization algorithms. Therefore, two basic noncontact measurement methods (sensor and satellite) were investigated in winter wheat, and their precision was evaluated in this study. Nitrogen uptake at four characteristic growth stages (BBCH 31, BBCH 39, BBCH 55, and BBCH 65) was determined using algorithms based on sensor and satellite data. As a reference, nitrogen uptake was determined using biomass samples in the laboratory (ground truth data). The precision of the tested methods was evaluated using statistical indicators (mean, median, minimum, maximum, and standard deviation) and correlation analyses between the nitrogen uptake of the ground truth data and that of the respective method. The results showed moderate to strong correlations with the nitrogen uptake of the ground truth data for both methods ($R^2 = 0.57$–0.83). Both sensor and satellite data best represented nitrogen uptake in BBCH 39 and 55 ($R^2 = 0.63$–0.83). In sum, there were only slight deviations in the absolute amount of nitrogen uptake ($\leq \pm 15\%$). Clear deviations can be explained by external influences during measurement. Overall, the investigations showed that the nitrogen uptake could be appropriately determined as a data basis for site-specific fertilization systems using sensor and satellite data.

**Keywords:** nitrogen uptake; sensor data; satellite data; site-specific fertilization; winter wheat




## 1. Introduction

Harmonizing successful crop production with environmental protection is a key requirement of modern fertilization systems. A particular focus is placed on nitrogen (N) fertilization. Nitrogen uptake by wheat in the field can vary noticeably. Spatial variability of nitrogen uptake depends on numerous overlapping influencing factors and their interactions (edaphic factors, climatic factors, and agricultural management practices) [1–6]. Particularly, soil properties, such as soil texture, available water capacity, humus content, nutrient content, and pH, vary on a very small scale, resulting in varying nitrogen uptakes [7–10]. This effect can be further intensified by prevalent uniform fertilization due to different nutrient removals in the high- and low-yield zones of a field [11–14]. This results in small-scale fluctuating nitrogen balances and stocks in soil, causing high nitrate leaching in low-yield zones with overfertilization [5,15]. Therefore, systems adapted to small-scale crop variations for fertilization will be required, which will consider the heterogeneity of fields and their different yield potentials to minimize nitrate losses.

Site-specific nitrogen fertilization is a promising approach to minimizing nitrate leaching [16–20]. The literature shows that this method can balance the nitrogen surplus and improve nitrogen efficiency [21–24]. In this context, various methods for site-specific nitrogen fertilization have been developed and tested. These approaches can be divided into

three categories: "mapping", "online", and "mapping + online" [16,25,26]. Site conditions (e.g., soil texture or yield potential) are used with the mapping approach, whereas crop biomass and/or nitrogen uptake are determined by field measurements (sensor/satellite) and algorithms in the online approach. The mapping + online approach is a combination of both. Fertilizer systems based on the mapping + online approach, which uses sensor or satellite data, have been established [16,22,23,27]. These systems use different methods to determine the nitrogen uptake of the crops in the field on a small scale. Based on the determined nitrogen uptake, these fertilizer systems calculate the amount of nitrogen fertilizer to be applied using algorithms and other data, such as the yield potential, quality target, or weather data [28,29]. Studies on this show that the accuracy of determining nitrogen uptake can vary significantly depending on the method [30,31]. For example, the use of different vegetation indices in reflection–optical measurements results in clear differences in nitrogen uptake [32–35]. The literature shows that some vegetation indices are more or less suitable for determining biomass growth and nitrogen uptake, whereas the suitability of other vegetation indices varies based on the crop's growth stage [36–38]. Studies on this show that the vegetation index red edge inflection point (REIP) can provide robust and accurate data on nitrogen uptake, particularly for winter wheat [4,39–41]. The precision of the determination is crucial since current nitrogen uptake is a significant parameter in fertilization algorithms. Deviations in the determination of nitrogen uptake lead to incorrect calculations, resulting in yield losses and environmental pollution [42,43]. Therefore, a precise evaluation of the most recent site-specific fertilization methods, particularly the determination of nitrogen uptake, with ground truth data is crucial for harmonizing successful crop production with environmental protection.

This study examines the accuracy of recording nitrogen uptake with two basic non-contact measurement methods of site-specific nitrogen fertilization in winter wheat. The aim was to evaluate their precision and suitability as important data for site-specific fertilization algorithms. Thus, plot trials were conducted in 2020 and 2021 at two different locations in southern Germany. The trials analyzed the accuracy of the individual methods in mapping the nitrogen uptake of winter wheat at different growth stages (BBCH 31, BBCH 39, BBCH 55, and BBCH 65) in a sub-area, as this is decisive for the precision of the fertilizer applications generated with fertilizer algorithms. Therefore, the following were investigated: (a) how accurately the relative differences in the field were identified by the methods and (b) how accurately the methods estimated absolute nitrogen uptake. In the trial plots, nitrogen uptake was determined using biomass samples (ground truth data) and digital, georeferenced methods (sensor and satellite). Correlation analyses evaluated the relationships among the nitrogen uptake data determined using different methods. Based on the results, the accuracy, precision, and suitability of the tested methods for recording the spatial variability of nitrogen uptake in winter wheat at different growth stages were evaluated.

## 2. Materials and Methods

### 2.1. Site and Weather Conditions

Two heterogeneous fields in which the experiments were conducted in 2020 and 2021 were selected. Both fields belong to the Makofen Research Farm (48°81′55″ N 12°74′31″ E), which is 15 km southeast of Straubing (320 m a.s.l.). The trial fields of the Makofen Research Farm are flat and characterized by extremely fertile loess soil. Table 1 shows the most important soil parameters in the trial fields.

Table 2 provides an overview of temperature and precipitation at Makofen Research Farm. The 20-year mean annual precipitation at the trial sites is 781 mm, and the mean annual temperature is 9.5 °C.

**Table 1.** Soil data—Makofen Research Farm.

| Property | Unit | Field A | Field B |
|---|---|---|---|
| Soil classification | | Cambisol | Cambisol |
| Soil type | | Silty loam | Silty loam |
| Soil fertility index * | | 75–85 | 70–80 |
| Sand (0–30 cm) | % | 6.0 | 6.9 |
| Silt (0–30 cm) | % | 70.1 | 69.4 |
| Clay (0–30 cm) | % | 23.9 | 23.7 |
| Available water capacity (in 10 cm) | Vol.% | 24.0 | 23.2 |
| Soil organic carbon content (0–30 cm) | % DM | 1.2 | 1.4 |
| Soil total nitrogen content (0–30 cm) | % DM | 0.14 | 0.12 |
| Plant available phosphorus content (0–30 cm) | mg $(100\ g)^{-1}$ | 14.8 | 17.9 |
| Plant available potassium content (0–30 cm) | mg $(100\ g)^{-1}$ | 17.7 | 18.4 |
| pH (0–30 cm) | | 6.5 | 6.9 |

* The soil fertility index is a quantitative assessment of soil fertility given in integers in a range of 0–100, with 100 representing the most fertile soil in Germany.

**Table 2.** Mean temperature and precipitation—Makofen Research Farm.

| | Unit | January to March | April to June | July to September | October to December | Year |
|---|---|---|---|---|---|---|
| **2000–2020 Makofen** | | | | | | |
| Temperature $\bar{x}$ | °C | 1.4 | 14.4 | 17.3 | 4.7 | 9.5 |
| Precipitation $\sum$ | mm | 170 | 209 | 230 | 172 | 781 |
| **2020 Makofen** | | | | | | |
| Temperature $\bar{x}$ | °C | 3.7 | 13.9 | 18.3 | 5.1 | 10.3 |
| Precipitation $\sum$ | mm | 149 | 189 | 176 | 141 | 655 |
| **2021 Makofen** | | | | | | |
| Temperature $\bar{x}$ | °C | 1.8 | 13.1 | 17.3 | 4.4 | 9.2 |
| Precipitation $\sum$ | mm | 129 | 268 | 250 | 165 | 812 |

*2.2. Crop Management*

In 2020 and 2021, the RGT Meister winter wheat variety was grown on the trial fields. The previous crop grown in the fields was sugar beets. Sowing, plant protection, and fertilization were conducted uniformly on the trial fields. Fertilization was conducted according to the Fertilizer Ordinance based on the Nmin content at the beginning of the spring growing season (2020: 66 kg N ha$^{-1}$; 2021: 62 kg N ha$^{-1}$). Plant protection was conducted according to the infestation situation. Table 3 shows an overview of crop management.

*2.3. Experimental Design*

The experimental setup was precisely adapted to the 10 m × 10 m grid of the satellite data. Both the plot size (10 m × 10 m) and the trial alignment in the field were based on the satellite data grid. This is critical for the high accuracy of the satellite data [5,44]. New plots were available for each growth stage, since the cutting of the biomass samples in the individual plots would influence the reflection measurements with the sensor and satellite at the subsequent growth stage. The experimental setup was the same in both experimental years, and only the number of plots differed (2020: *n* = 30; 2021: *n* = 45). Figure 1 shows the experimental setup in 2020.

*2.4. Methods of Determining Nitrogen Uptake*

Nitrogen uptake per plot was determined using the following methods:

- Biomass samples (ground truth data) [45,46];
- An algorithm based on reflection measurements using a multispectral sensor [28,47];
- Radiative transfer model (soil–leaf–canopy) based on satellite data [29,48].

**Table 3.** Crop management of the trial fields.

| Field | Treatment | Unit | Amount | Product | Date |
|---|---|---|---|---|---|
| A | Sowing | kg/ha$^{-1}$ | 156 | Meister | 27 October 2019 |
| A | First N fertilization | kg/ha$^{-1}$ | 60 | ASN | 28 March 2020 |
| A | Second N fertilization | kg/ha$^{-1}$ | 80 | CAN | 30 April 2020 |
| A | Third N fertilization | kg/ha$^{-1}$ | 40 | CAN | 20 May 2020 |
| A | N fertilization, total | kg/ha$^{-1}$ | 180 | | |
| A | Plant protection | kg/ha$^{-1}$ | 0.05/0.07 | Biathlon, Concert | 7 April 2020 |
| A | Plant protection | L/ha$^{-1}$ | 0.5 | CCC 720 | 7 April 2020 |
| A | Plant protection | L/ha$^{-1}$ | 1.25/0.075 | Capalo/Karate | 16 May 2020 |
| A | Plant protection | L/ha$^{-1}$ | 2.0 | Osiris | 13 June 2020 |
| B | Sowing | kg/ha$^{-1}$ | 205 | Meister | 10 November 2020 |
| B | First N fertilization | kg/ha$^{-1}$ | 78 | ASN | 4 March 2021 |
| B | Second N fertilization | kg/ha$^{-1}$ | 54 | CAN | 8 May 2021 |
| B | Third N fertilization | kg/ha$^{-1}$ | 40 | CAN | 4 June 2021 |
| B | N fertilization, total | kg/ha$^{-1}$ | 172 | | |
| B | Plant protection | kg/ha$^{-1}$ | 0.13 | Broadway | 22 April 2021 |
| B | Plant protection | L/ha$^{-1}$ | 0.25/0.5 | Pixxaro/CCC 720 | 22 April 2021 |
| B | Plant protection | L/ha$^{-1}$ | 1.0/0.3 | Revystar/Flexity | 20 May 2021 |
| B | Plant protection | L/ha$^{-1}$ | 1.0/0.075 | Ascra Xpro/Karate | 11 June 2021 |

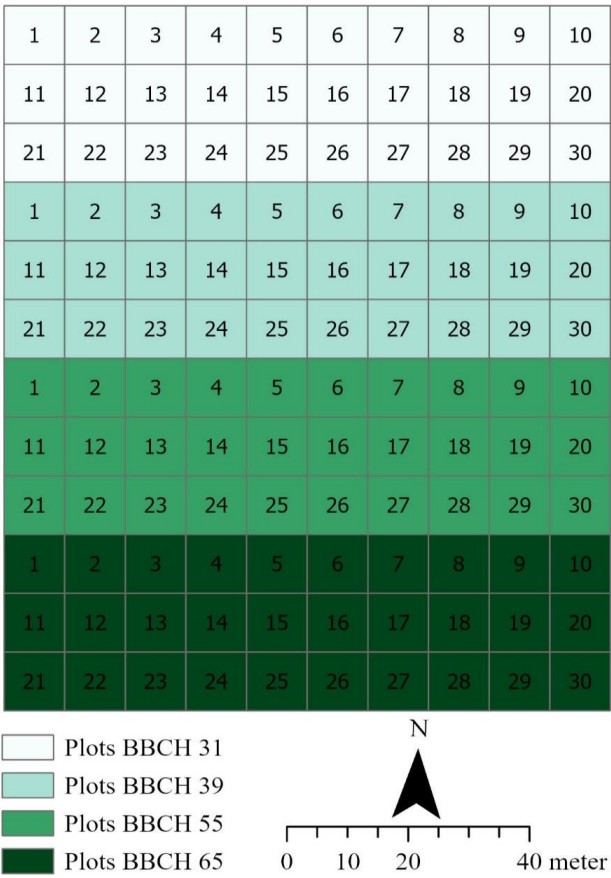

**Figure 1.** Experimental setup (Field A, 2020).

Nitrogen uptake was determined using the respective methods in the growth stages, BBCH 31, BBCH 39, BBCH 55, and BBCH 65. Thus, an area of 2.5 m$^2$ of plants was manually cut off in each plot for the ground truth data. These samples were weighed, chopped, and dried at 105 °C. This resulted in the above-ground biomass yield. The Dumas method was used to analyze the nitrogen content of the samples, and the nitrogen uptake of the plot

was determined [45,46]. The reflection measurements with the multispectral sensor were conducted in the respective growth stages in the individual plots. The REIP 700 vegetation index was calculated based on these measurements, and the algorithms used to estimate the nitrogen uptake were based on this index considering further data, such as yield potential and variety properties [28]. Depending on the availability, the radiative transfer model used up-to-date satellite data to estimate nitrogen uptake. Based on the satellite data, the radiative transfer model calculated the nitrogen uptake at the respective growth stages considering additional data, such as observational parameters, soil reflectance information, leaf optical properties, and canopy properties [29,48].

*2.5. Data Processing*

Considering the corresponding methodology, nitrogen uptake was determined using different methods. Point data were generated using digital contactless measuring systems. Next, these point data were visualized using geoinformation system software, ArcGIS [49], and assigned to the digitized plots via their coordinates. Data points on or outside the plot edges were removed. The recorded data points varied in spatial resolution and distribution based on the method. Figure 2 shows the detailed structure and data distribution of a plot. Subsequently, the mean was calculated using all available data points per plot and method. Thus, the nitrogen uptake per plot in kg N ha$^{-1}$ was determined for each method.

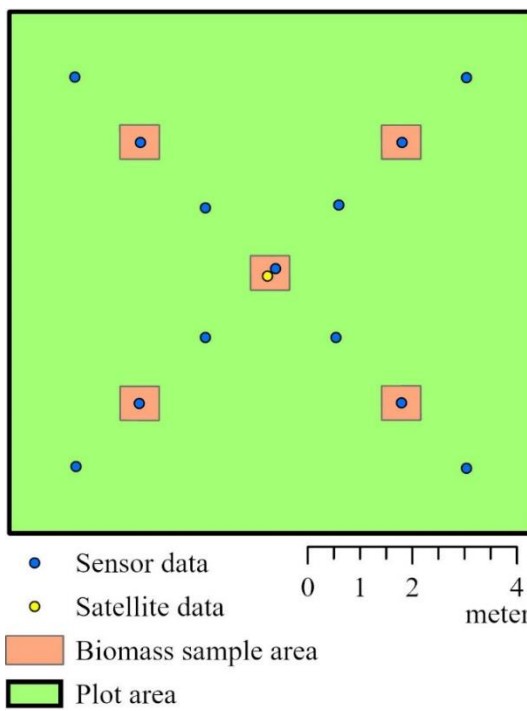

**Figure 2.** Structure and data distribution in the plot.

*2.6. Descriptive Statistics*

The mean, median, minimum, maximum, and standard deviation were calculated for each method using R.

*2.7. Correlation Analysis*

Correlation analyses based on the nitrogen uptake per plot in kg N ha$^{-1}$ determined the relationships between the data of the tested digital methods and the ground truth data. The coefficients of determination ($R^2$) were classified as very strong ($R^2 > 0.9$), strong ($0.9 > R^2 > 0.7$), moderate ($0.7 > R^2 > 0.5$), weak ($0.5 > R^2 > 0.3$), or very weak ($R^2 < 0.3$).

## 3. Results

### 3.1. Spatial Variation in Nitrogen Uptake in 2020 (Field A)

Different methods for the site-specific determination of nitrogen uptake at characteristic growth stages produced different results in the nitrogen distribution pattern, nitrogen variation, and mean nitrogen uptake in Field A (Figure 3, Table 4). The nitrogen uptake of the biomass samples (ground truth data) in BBCH 31 varied between 33.2 and 64.1 kg N ha$^{-1}$. The nitrogen uptake estimated by the radiative transfer model based on satellite data in BBCH 31 (23.3–35.8 kg N ha$^{-1}$) was also characterized by variability; however, the variation was not as great as in those obtained with the other methods, and a significantly lower nitrogen level was noticeable. The nitrogen uptake estimated using algorithms based on sensor data in BBCH 31 (24.6–66.2 kg N ha$^{-1}$) was more similar to the measured values of the ground truth data. All the methods in BBCH 39 showed almost the same mean nitrogen uptake and a similar nitrogen distribution, but the variation was higher for both satellite and sensor data (Figure 3).

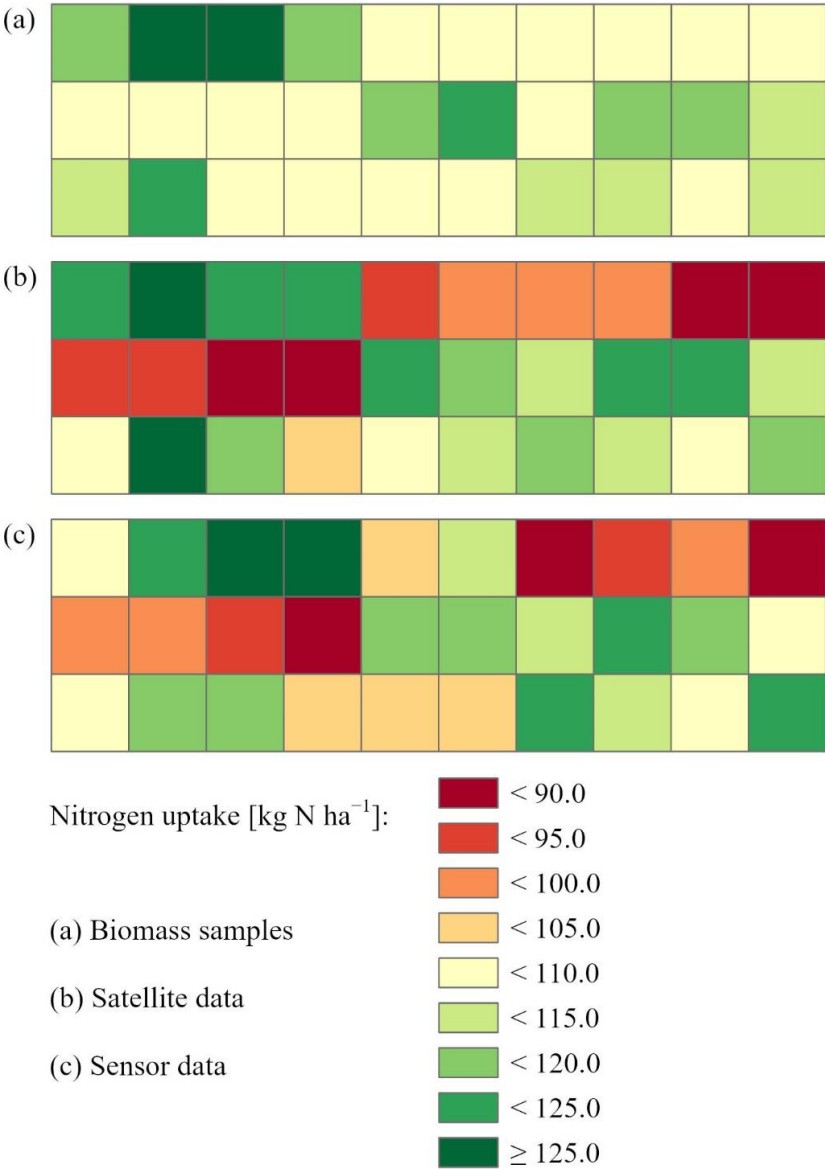

**Figure 3.** Nitrogen uptake 2020, BBCH 39. Nitrogen uptake was determined from the biomass samples, sensor data, and satellite data.

**Table 4.** Descriptive statistics of the nitrogen uptake data in kg N ha$^{-1}$ analyzed in this study.

| Variable | n | Year | BBCH | Unit | Mean | Median | Minimum | Maximum | Standard Deviation | Skewness |
|---|---|---|---|---|---|---|---|---|---|---|
| Biomass samples | 30 | 2020 | 31 | kg N ha$^{-1}$ | 50.2 | 50.9 | 33.2 | 64.1 | 7.9 | −0.41 |
| Satellite data | 30 | 2020 | 31 | kg N ha$^{-1}$ | 30.4 | 30.7 | 23.3 | 35.8 | 3.4 | −0.36 |
| Sensor data | 30 | 2020 | 31 | kg N ha$^{-1}$ | 42.7 | 43.7 | 24.6 | 66.2 | 9.8 | 0.28 |
| Biomass samples | 30 | 2020 | 39 | kg N ha$^{-1}$ | 118.2 | 118.4 | 109.2 | 125.2 | 3.7 | −0.34 |
| Satellite data | 30 | 2020 | 39 | kg N ha$^{-1}$ | 116.9 | 122.7 | 84.9 | 141.6 | 17.2 | −0.51 |
| Sensor data | 30 | 2020 | 39 | kg N ha$^{-1}$ | 124.1 | 127.1 | 68.8 | 169.1 | 27.2 | −0.37 |
| Biomass samples | 30 | 2020 | 55 | kg N ha$^{-1}$ | 186.8 | 187.9 | 167.1 | 199.5 | 9.6 | −0.45 |
| Satellite data | 30 | 2020 | 55 | kg N ha$^{-1}$ | 144.9 | 145.4 | 121.6 | 163.0 | 11.9 | −0.34 |
| Sensor data | 30 | 2020 | 55 | kg N ha$^{-1}$ | 203.6 | 216.1 | 143.5 | 247.3 | 32.2 | −0.52 |
| Biomass samples | 30 | 2020 | 65 | kg N ha$^{-1}$ | 225.5 | 225.9 | 211.8 | 235.2 | 6.1 | −0.37 |
| Satellite data | 30 | 2020 | 65 | kg N ha$^{-1}$ | 178.1 | 181.2 | 164.1 | 188.3 | 7.9 | −0.57 |
| Sensor data | 30 | 2020 | 65 | kg N ha$^{-1}$ | 248.5 | 255.3 | 166.9 | 320.2 | 37.9 | −0.30 |
| Biomass samples | 45 | 2021 | 31 | kg N ha$^{-1}$ | 45.2 | 44.9 | 29.4 | 63.0 | 7.5 | 0.12 |
| Satellite data | 45 | 2021 | 31 | kg N ha$^{-1}$ | 40.8 | 40.5 | 33.8 | 47.6 | 1.9 | −0.11 |
| Sensor data | 45 | 2021 | 31 | kg N ha$^{-1}$ | 43.9 | 44.9 | 23.5 | 62.1 | 9.7 | −0.34 |
| Biomass samples | 45 | 2021 | 39 | kg N ha$^{-1}$ | 144.3 | 142.2 | 124.1 | 195.8 | 13.9 | 1.17 |
| Satellite data | 45 | 2021 | 39 | kg N ha$^{-1}$ | 123.4 | 120.4 | 100.8 | 161.0 | 16.0 | 0.9 |
| Sensor data | 45 | 2021 | 39 | kg N ha$^{-1}$ | 143.0 | 133.7 | 103.8 | 217.5 | 31.2 | 0.51 |
| Biomass samples | 45 | 2021 | 55 | kg N ha$^{-1}$ | 192.3 | 192.1 | 142.6 | 225.9 | 16.1 | −0.74 |
| Satellite data | 45 | 2021 | 55 | kg N ha$^{-1}$ | 170.0 | 169.2 | 146.4 | 202.8 | 12.5 | 0.47 |
| Sensor data | 45 | 2021 | 55 | kg N ha$^{-1}$ | 199.9 | 191.6 | 118.3 | 275.7 | 44.1 | 0.24 |
| Biomass samples | 45 | 2021 | 65 | kg N ha$^{-1}$ | 218.3 | 217.5 | 182.4 | 260.8 | 17.5 | 0.52 |
| Satellite data | 45 | 2021 | 65 | kg N ha$^{-1}$ | 183.4 | 182.6 | 140.8 | 225.5 | 21.4 | 0.19 |
| Sensor data | 45 | 2021 | 65 | kg N ha$^{-1}$ | 232.1 | 239.3 | 147.2 | 308.5 | 46.9 | 0.11 |

Both the satellite and sensor data in BBCH 55 and 65 showed similar nitrogen distributions, which was consistent with the ground truth data. However, the absolute level of nitrogen uptake was noticeably lower with the satellite data than with the ground truth data, whereas it was higher with the sensor data. A deviation of −20% in the mean nitrogen uptake with the satellite data and +10% with the sensor data was observed in BBCH 55 and 65.

### 3.2. Spatial Variation in Nitrogen Uptake in 2021 (Field B)

The nitrogen uptake data determined using different digital measuring systems and methods for the examined growth stages in Field B produced results similar to those in Field A (Figure 4, Table 4). Thus, the nitrogen uptake of the ground truth data in BBCH 31 (29.4–63.0 kg N ha$^{-1}$) varied in a similar range as in 2020. The estimate of nitrogen uptake by the radiative transfer model based on satellite data in BBCH 31 (33.8–47.6 kg N ha$^{-1}$) was also characterized by variability, but the variation was again lower and at a lower nitrogen level, whereas the estimate from the sensor data (23.5–62.1 kg N ha$^{-1}$) was similar to the measured values of the ground truth data (Figure 4). All methods in BBCH 39 showed a similar nitrogen distribution; however, the variation was slightly lower with the satellite data and was slightly higher with the sensor data. The satellite and sensor data in BBCH 55 and 65 showed similar nitrogen distribution patterns, which were consistent with the ground truth data. However, it was also noticed that the absolute level of nitrogen uptake was lower with the satellite data than with the ground truth data, whereas it was slightly higher with the sensor data. In BBCH 55 and 65, there were slight deviations from the mean nitrogen uptake of −14% with the satellite data and +5% with the sensor data.

### 3.3. Correlation between Variables

Table 5 shows the coefficients of determination ($R^2$) of the linear relationships of the nitrogen uptake data determined using various digital methods.

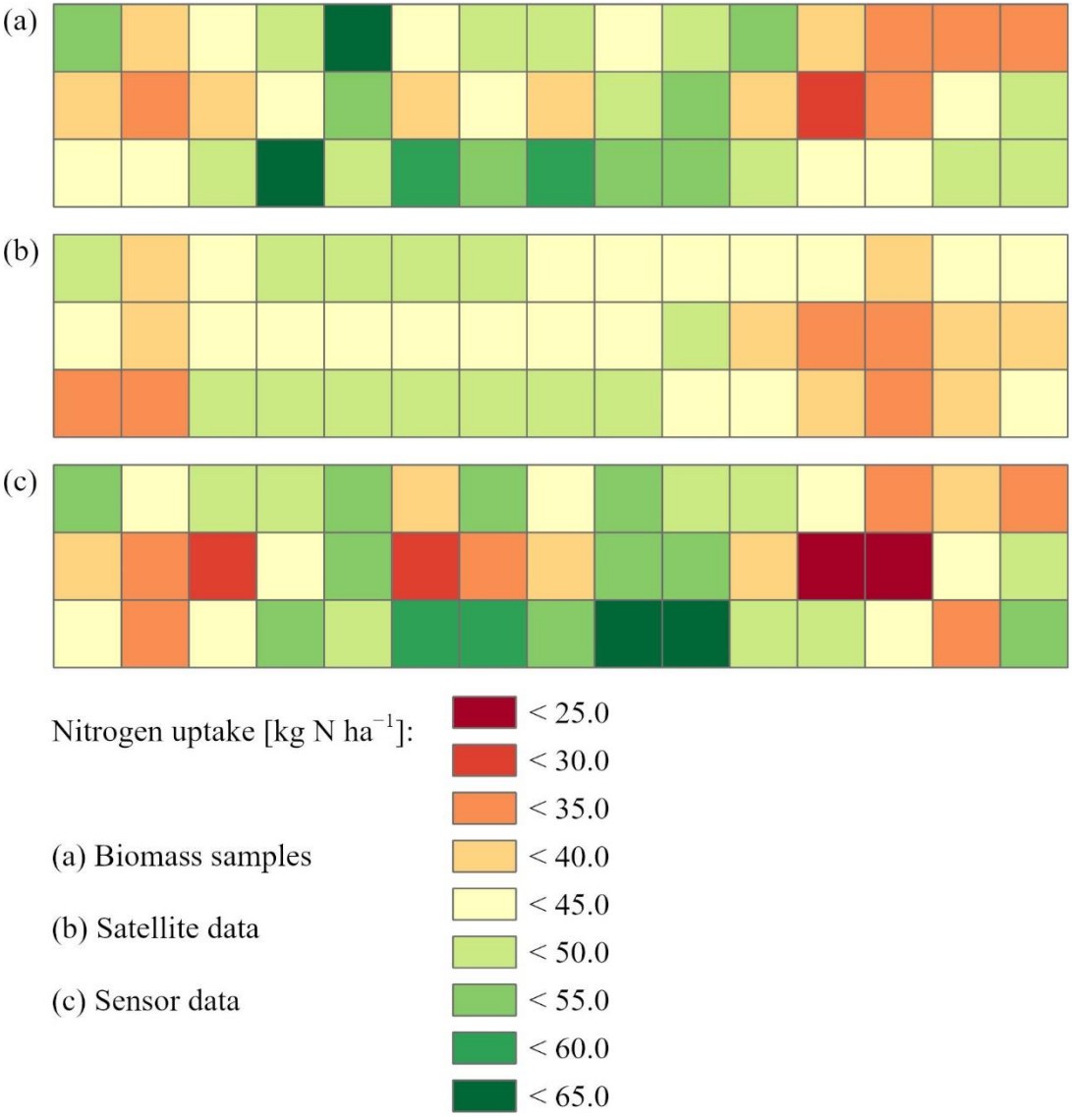

**Figure 4.** Nitrogen uptake 2021, BBCH 31. Nitrogen uptake was determined from the biomass samples, sensor data, and satellite data.

**Table 5.** Coefficients of determination ($R^2$): nitrogen uptake data for 2020 (*n* = 30) and 2021 (*n* = 45).

| $R^2$ | BBCH | Sensor 2020 | Satellite 2020 | Sensor 2021 | Satellite 2021 |
|---|---|---|---|---|---|
| Biomass samples 2020 | 31 | 0.74 | 0.60 | | |
| Biomass samples 2020 | 39 | 0.83 | 0.80 | | |
| Biomass samples 2020 | 55 | 0.77 | 0.74 | | |
| Biomass samples 2020 | 65 | 0.67 | 0.67 | | |
| Biomass samples 2021 | 31 | | | 0.66 | 0.48 |
| Biomass samples 2021 | 39 | | | 0.76 | 0.57 |
| Biomass samples 2021 | 55 | | | 0.72 | 0.63 |
| Biomass samples 2021 | 65 | | | 0.65 | 0.59 |

### 3.3.1. Field A (2020)

Overall, all tested methods achieved similar correlations and could at least moderately map the nitrogen uptake for all growth stages examined. In BBCH 31, the correlation analysis showed a strong relationship between the ground truth data and the estimate from the sensor data ($R^2 = 0.74$). The nitrogen uptake in BBCH 31 determined by the radiative transfer model based on satellite data ($R^2 = 0.60$) was moderately correlated with the ground truth data. In BBCH 39 and 55, both the estimates from the sensor data ($R^2 = 0.77–0.83$) and those from the satellite data ($R^2 = 0.74–0.80$) were strongly correlated with the ground truth data (Figure 5). The results from the sensor and satellite data in BBCH 65 were identical and showed a moderate correlation with the ground truth data ($R^2 = 0.67$).

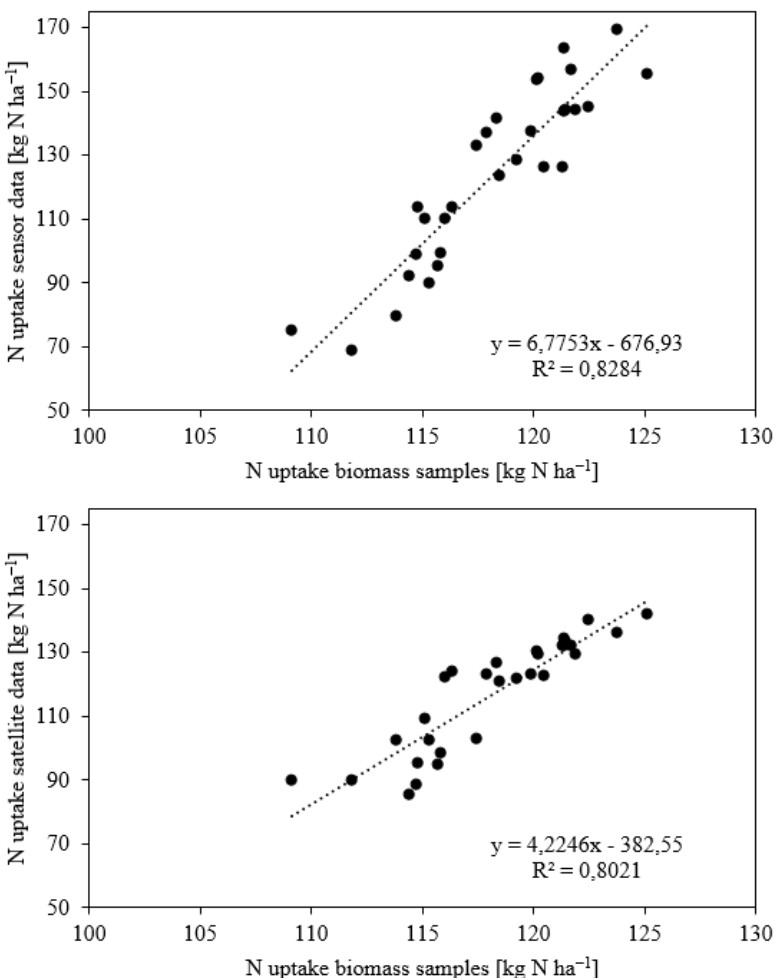

**Figure 5.** The linear relationships between nitrogen uptake of ground truth data and sensor data (**above**) and satellite data (**below**) in BBCH 39 in Field A in 2020.

### 3.3.2. Field B (2021)

In 2021, the correlations were considerably similar to those of the previous year. All methods except the satellite data in BBCH 31 mapped nitrogen uptake at least moderately. There was a moderate correlation between the ground truth data and the sensor data ($R^2 = 0.66$) in BBCH 31, but only a weak correlation with the satellite data ($R^2 = 0.48$). In BBCH 39 and 55, the estimates from the sensor data ($R^2 = 0.72–0.76$) were strongly correlated with the ground truth data, and with those from the satellite data ($R^2 = 0.57–0.63$) moderately. The results of the sensor ($R^2 = 0.65$) and satellite data ($R^2 = 0.59$) in BBCH 65 were repeatedly similar and showed a moderate correlation with the ground truth data.

## 4. Discussion

### *4.1. Discussion of the Methods*

This study investigated the recording of nitrogen uptake using two basic noncontact measurement methods of site-specific nitrogen fertilization in winter wheat at characteristic growth stages in heterogeneous fields at two locations in southern Germany. The precision of the methods was tested by comparing the statistical indicators (mean, median, minimum, maximum, and standard deviation) and examining correlative relationships. The aim of this was to identify the variability in nitrogen uptake and analyze its absolute amount.

#### 4.1.1. Site Selection

The expression of the spatial variability of nitrogen uptake has a significant impact on the results of this method's comparison [50,51]. Therefore, heterogeneous fields were selected for this study. The heterogeneity of the trial sites was assessed based on the soil parameters, biomass maps, and a farm manager's expertise. Heijting et al. [52] showed that a farm manager's expertise is a suitable basis for evaluating the heterogeneity of a field. Furthermore, other studies revealed that small-scale variations in the soil properties and crop stands are characteristic of the study region [5,53].

#### 4.1.2. Ground Truth Data

The nitrogen uptake data were determined using two digital methods. Suitable ground truth data (biomass samples) are crucial in evaluating the different estimation methods for comparing the modeled data with the measured data [5]. Therefore, in this study, biomass samples were cut in all plots for each examined growth stage (BBCH 31, BBCH 39, BBCH 55, and BBCH 65), and the nitrogen uptake was determined in the laboratory [45,46]. The biomass samples enabled accurate determination of the nitrogen uptake per plot and the evaluation of the estimates obtained using digital methods. However, the measuring effort for biomass samples is extremely high and a limiting factor for large areas. For example, Mittermayer et al. [5] investigated the variability of nitrogen uptake in an area of 13.1 ha and used 50 biomass samples; the data analysis was conducted using geostatistical methods. Other studies analyzed even larger fields of more than 1000 ha and only compared the sensor and satellite data. Mezera et al. [51] and Gozdowski et al. [54] achieved similar results with both measurement methods and found moderate to strong correlations ($R^2 = 0.51$–$0.79$). The results of this study confirmed this. For example, both methods correlated strongly ($R^2 = 0.76$) in BBCH 39 in 2020; however, there was a higher variation in absolute nitrogen uptake with both digital methods than with the ground truth data. Because the absolute height of nitrogen uptake is also crucial for site-specific fertilization, the ground truth data of the biomass samples were of immense importance for the precise evaluation of the two digital methods in this study.

### *4.2. Discussion of the Results*

#### 4.2.1. Sensor Data

The nitrogen uptake estimate based on multispectral sensor data, the REIP vegetation index, and a crop-specific algorithm [28] provided reliable results in both test years. The method recognized the spatial nitrogen distribution in all tested growth stages as moderate to strong ($R^2 = 0.65$–$0.83$). Apart from BBCH 31 in 2020 ($-15\%$), there were only small deviations ($\leq \pm 10\%$) in the nitrogen uptake's mean absolute level. This deviation may have been due to drought stress. There was a pronounced early summer drought at the time of the measurements in BBCH 31. Drought stress, plant diseases, soil compaction, and lack of other nutrients can influence the reflection signature in reflection–optical measurements, resulting in incorrectly interpreted measured values [55–57]. Further, the correlation quality of the sensor data typically improved with the increasing growth stage toward a peak in BBCH 39 and then slightly decreased again. Nevertheless, no clear saturation occurred, as, for example, with systems based on simple vegetation indices, such as NDVI or SAVI, and good precision was shown even with high nitrogen uptake [36,38,58]. Similar results

were also obtained by Prey and Schmidhalter [38,58], who investigated the sensitivity of different vegetation indices for estimating nitrogen uptake in winter wheat and consistently achieved moderate results with the REIP ($R^2$ = 0.59). Westermeier and Maidl [36] made the same conclusions in their study and even found correlations of up to $R^2$ = 0.9 for the REIP 700 index. Further investigations with reflection–optical sensor measurements presented moderate to very strong correlations with nitrogen uptake in winter wheat at the relevant growth stages ($R^2$ = 0.57–0.89) [4,5,59,60]. Consequently, the sensor data are suitable for both early and late site-specific fertilization measures. In addition, the sensor measurements from BBCH 65 can be used to calculate yield estimates and yield potential maps [61,62]. These results confirm the significant potential of modern sensor technology for recording nitrogen uptake as a basis for site-specific fertilization. The prerequisites for the successful implementation of this method are multispectral sensors with high measurement accuracy, suitable vegetation indices, and science-based algorithms [5,36,38,58,60].

### 4.2.2. Satellite Data

The nitrogen uptake estimate with the radiative transfer model based on satellite data also achieved good results in both test years. The method identified the spatial nitrogen distribution in all tested growth stages, except in BBCH 31 in 2021 ($R^2$ = 0.48), moderately to strongly ($R^2$ = 0.57–0.80). This deviation can be explained by the dependence of this method on clear, cloud-free satellite images [63–65]. In BBCH 31, it was frequently cloudy in 2021, and the availability of cloud-free satellite images was extremely limited. Consequently, older satellite images had to be used to estimate nitrogen uptake, which can result in deviations. The mean absolute level of nitrogen uptake of the satellite data in the years 2020 ($\leq -20\%$) and 2021 ($\leq -15\%$) showed slightly larger deviations compared to those with the sensor data. An exception to this was BBCH 31 in 2020 ($-40\%$), since there was a significant deviation, which can be explained by drought stress. Drought stress, plant diseases, soil compaction, and a lack of other nutrients can affect the reflection signature of multispectral satellite images in the same way as with the sensor data, resulting in incorrectly interpreted measured values [55,56]. Other literature also presented good results using satellite data. Chen [66] conducted correlation analyses between remote sensing data and the nitrogen concentration in winter wheat at different growth stages and achieved strong correlations ($R^2$ = 0.86). Magney et al. [67] compared satellite data with biomass samples and successfully mapped the nitrogen uptake with high precision ($R^2$ = 0.81). Further investigations into mapping the nitrogen uptake of winter wheat using satellite data also showed good results ($R^2$ = 0.74) [68]. Consequently, with current data availability, satellite data are also suitable as a basis for early and late site-specific fertilization measures. Further, yield estimates and yield potential maps can also be generated using satellite images [29,69]. These results confirm that remote sensing methods can be used to record parameters such as nitrogen uptake with good precision and use them for site-specific fertilization measures.

### 5. Conclusions

Current nitrogen uptake is a crucial parameter in site-specific fertilization algorithms. The more precisely the nitrogen uptake is determined by noncontact measuring methods, the more precise the result of the site-specific fertilization. The results of these investigations show the suitability of both measurement methods. Nitrogen uptake can be determined appropriately using the tested methods for both BBCH 31 and 39, which are crucial growth stages for yield fertilization, and BBCH 55 and 65, which are relevant for quality fertilization or for deriving yield estimates. Significant deviations, such as those in BBCH 31 in 2020, can be explained by external influences. Further, data generated by sensor measurements close to plants are somewhat more precise, particularly when determining the absolute level of nitrogen uptake. In addition, the sensor technology is unaffected by cloud cover and is particularly superior at times when there are no cloud-free satellite images. On the other hand, the sensor technology is extremely expensive and requires a high level of

user qualification. However, with the satellite data, the finished application map is sent to the machine and is, therefore, very easy to use. Furthermore, the satellite data depict the entire field and do not only measure partial areas. In summary, both measurement methods have advantages and disadvantages. However, both methods prove their potential and are suitable for determining the nitrogen uptake for site-specific fertilization systems in winter wheat. Referring to the great relevance of the topic and the environmental effects of inappropriate fertilization, noncontact measuring methods for determining plant parameters such as nitrogen uptake require urgent further investigation to improve the precision, particularly with the absolute level of nitrogen uptake. The focus should be on the higher spatial resolution of satellite data (e.g., 5 m × 5 m) and other wavelengths in reflection–optical measurements to improve and develop vegetation indices.

**Author Contributions:** Conceptualization, M.S. and F.-X.M.; methodology, M.S. and F.-X.M.; investigation, M.S.; data curation, M.S.; writing—original draft preparation, M.S.; writing—review and editing, F.-X.M., K.-J.H. and H.B.; project administration, J.S.; funding acquisition, J.S. All authors have read and agreed to the published version of the manuscript.

**Funding:** This research was funded by the European Commission and the State of Bavaria as part of EIP-Agri (EP4-904).

**Data Availability Statement:** The data presented in this study are available from the corresponding author on reasonable request.

**Acknowledgments:** We would like to thank VISTA Geowissenschaftliche Fernerkundung GmbH (Gabelsbergerstraße 51, 80333 Munich, Germany) for providing nitrogen uptake data using satellite data and a radiative transfer model.

**Conflicts of Interest:** The authors declare no conflict of interests.

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
