# Peer review of "Analysis of Nitrogen Uptake in Winter Wheat Using Sensor and Satellite Data for Site-Specific Fertilization"

_agronomy, doi:10.3390/agronomy12061455_

Round 1

Reviewer 1 Report

The work entitled "Determination of nitrogen uptake by winter wheat using sensor and satellite data for site-specific fertilization" concerns two basic non-contact measurement methods (sensor and satellite) which were studied in winter wheat and their precision. It was very well stated that this study is acceptable with a slight revision and suitable for publication in this journal. The manuscript is clearly and intelligible. The results are interesting and can be used in further work. I'm agree with the author that other literature also presented good results using satellite date. Despite minor editorial and linguistic improvements, the manuscript may be acceptable for the journal.

Author Response

Dear Reviewer 1,

first of all, thank you for your comments. I have made a slight revision and incorporated all the editorial improvements. 

Reviewer 2 Report

Please find in the following my comments about the review of a manuscript under the title (Determination of Nitrogen Uptake in Winter Wheat Using Sensor and Satellite Data for Site-Specific Fertilization):

In this study, the authors investigated two basic noncontact measurement methods (sensor and satellite) in winter wheat, and their precision is evaluated

Main question addressed by the research

This study focused on the examination of the accuracy of recording the nitrogen uptake of two basic non-contact measurement methods of site-specific nitrogen fertilization in winter wheat.

Originality and relevance

§  The study is interesting for reading and relevant in the field as it evaluated two basic non-contact measurement methods precision and suitability as important data for site-specific fertilization algorithms

§  The study has moderate scientific quality.

§  The study is relevant to the scope of this journal.

§  The manuscript is clear, and relevant for the field and its presentation needs fine modifications.

Comments:

Results:

§  The title of the table and legends of the figures should be informative and self-explanatory? Revise.

Author Response

Dear Reviewer 2,

first of all, thank you for your comment, which helps to improve our article.

I have revised the titles. 

Reviewer 3 Report

Overall, the manuscript has been written well, also has a new information and this kind of study added value to the journal. Just in conclusions the authors has been wrote references why? I think if better to focus in your results and discussion

Author Response

Dear Reviewer 3,

first of all, thank you for your comment, which helps to improve our article.

I have revised the conclusions and removed the references in this chapter.